# Plant diversity maintains multiple soil functions in future environments

**Nico Eisenhauer[1,2†]\*, Jes Hines[1,2†], Forest Isbell[3], Fons van der Plas[4], Sarah E Hobbie[3], Clare E Kazanski[3], Anika Lehmann[5,6], Mengyun Liu[1,2,7,8], Alfred Lochner[1,2], Matthias C Rillig[5,6], Anja Vogel[1,2,9], Kally Worm[10], Peter B Reich[10,11]**

[1]German Centre for Integrative Biodiversity Research Halle-Jena-Leipzig, Leipzig, Germany; [2]Institute of Biology, Leipzig University, Leipzig, Germany; [3]Department of Ecology, Evolution, and Behavior, University of Minnesota, St Paul, United States; [4]Department of Systematic Botany and Functional Biodiversity, Leipzig University, Leipzig, Germany; [5]Institute of Biology, Freie Universität Berlin, Berlin, Germany; [6]Berlin-Brandenburg Institute of Advanced Biodiversity Research, Berlin, Germany; [7]Key Laboratory of Vegetation and Management of Degraded Ecosystems, South China Botanical Garden, Chinese Academy of Sciences, Guangzhou, China; [8]University of Chinese Academy of Sciences, Beijing, China; [9]Institute of Ecology and Evolution, Friedrich Schiller University Jena, Jena, Germany; [10]Department of Forest Resources, University of Minnesota, St Paul, United States; [11]Hawkesbury Institute for the Environment, Western Sydney University, Sydney, Australia

**Abstract** Biodiversity increases ecosystem functions underpinning a suite of services valued by society, including services provided by soils. To test whether, and how, future environments alter the relationship between biodiversity and multiple ecosystem functions, we measured grassland plant diversity effects on single soil functions and ecosystem multifunctionality, and compared relationships in four environments: ambient conditions, elevated atmospheric $CO_2$, enriched N supply, and elevated $CO_2$ and N in combination. Our results showed that plant diversity increased three out of four soil functions and, consequently, ecosystem multifunctionality. Remarkably, biodiversity-ecosystem function relationships were similarly significant under current and future environmental conditions, yet weaker with enriched N supply. Structural equation models revealed that plant diversity enhanced ecosystem multifunctionality by increasing plant community functional diversity, and the even provision of multiple functions. Conserving local plant diversity is therefore a robust strategy to maintain multiple valuable ecosystem services in both present and future environmental conditions.
DOI: https://doi.org/10.7554/eLife.41228.001

**\*For correspondence:**
nico.eisenhauer@idiv.de

†These authors contributed equally to this work

**Competing interests:** The authors declare that no competing interests exist.

## Introduction

Many experimental studies have shown that both the average levels (*Hooper et al., 2005*; *Cardinale et al., 2012*; *Duffy et al., 2017*) and temporal stability (*Isbell et al., 2015*) of ecosystem functions increase with biodiversity. While the generality of positive relationships between biodiversity and ecosystem functioning is well established (*Cardinale et al., 2012*; *Duffy et al., 2017*; *Isbell et al., 2015*; *Lefcheck et al., 2015*), current research focuses on the underlying mechanisms (*Eisenhauer et al., 2016*; *Zuppinger-Dingley et al., 2014*; *Laforest-Lapointe et al., 2017*) and context-dependencies (*Craven et al., 2016*; *Guerrero-Ramírez et al., 2017*) of biodiversity-ecosystem function relationships, given that ecosystems face progressive environmental changes (*Barros and*

*IPCC. Climate Change, 2014*). One of the most prominent explanations of why diverse communities perform better than simple ones is that different species complement or facilitate each other in chemical, spatial, and temporal resource use (*Loreau and Hector, 2001*; *Eisenhauer, 2012*), resulting in higher functional diversity. Alternatively, diverse communities may have higher functioning because of an elevated probability of containing and becoming dominated by one or a few productive species (*Loreau and Hector, 2001*). Moreover, there is growing evidence that biotic interactions across trophic levels are important drivers of biodiversity effects (*Maron et al., 2011*; *Schnitzer et al., 2011*; *Eisenhauer et al., 2012a*) and that these interactions are significantly modulated by environmental conditions (*Guerrero-Ramírez and Eisenhauer, 2017*). As a consequence, biodiversity effects may depend on the environmental context, such as climatic conditions and resource availability (*Craven et al., 2016*; *Guerrero-Ramírez et al., 2017*), and thus may change in future environments. However, studies testing biodiversity-ecosystem function relationships under future conditions are rare (*Reich et al., 2001*; *He et al., 2002*; *Hooper et al., 2012*; *Thakur et al., 2015*), and typically investigate only one or a few ecosystem functions.

Soils provide many ecosystem functions (*Bardgett and van der Putten, 2014*) that contribute to human well-being, including resources that support plant biomass production, such as nutrients and water, soil carbon storage, and soil erosion control (*Wall et al., 2015*; *Amundson et al., 2015*). Consequently, concern about factors that influence soil functioning have attracted increasing scientific (*Bardgett and van der Putten, 2014*; *Amundson et al., 2015*; *Veresoglou et al., 2015*) and public attention (*Wall et al., 2015*; *World Health Organization and Secretariat of the Convention on Biological Diversity, 2014*). Changes in plant diversity have been identified as a key factor influencing soil organisms (*Hooper et al., 2000*; *Scherber et al., 2010*; *Eisenhauer et al., 2013*; *Lange et al., 2015*). Through bottom-up effects, diverse plant communities provide a higher quantity and quality of plant-derived inputs to soil microorganisms and detritivores (*Hooper et al., 2000*; *Eisenhauer et al., 2013*), which has cascading effects on the abundance and diversity at higher trophic levels in the soil (*Hooper et al., 2000*; *Scherber et al., 2010*; *Eisenhauer et al., 2013*). Ecosystem functions mediated by soil organisms, such as soil carbon storage (*Lange et al., 2015*; *Fornara and Tilman, 2008*), litter decomposition (*Vogel et al., 2013*), and soil aggregate stabilization (*Gould et al., 2016*), are thus sensitive to changes in the diversity of plant communities. Given the ecological and economic importance of soil functions (*Bardgett and van der Putten, 2014*; *Wall et al., 2015*; *Amundson et al., 2015*), a key research priority is to understand whether maintaining high local plant diversity secures multiple soil-mediated functions in future environmental conditions.

Ecosystem functioning is inherently multidimensional, and so multifunctionality measures have been increasingly used to summarize the ability of an ecosystem to deliver multiple functions or services simultaneously (*Manning et al., 2018*). Future environmental conditions may alter the relationship between biodiversity and ecosystem multifunctionality (*Manning et al., 2018*; *Hector and Bagchi, 2007*; *Soliveres et al., 2016*; *Byrnes et al., 2014*), potentially in distinct ways depending on the response of plant species abundances, or evenness, to new conditions. There is considerable evidence that different plant species promote different ecosystem functions (*Hector and Bagchi, 2007*), and that different plant species provide different ecosystem functions under different global change scenarios (*Isbell et al., 2011*). Resource additions often favor 'fast' (i.e. acquisitive) plant strategies, thereby decreasing the evenness, functional diversity, and species richness of plant communities (*Reich, 2014a*). In some cases, the contribution of those fast species to multiple ecosystem functions might not be strong enough to compensate for losses in contribution of sub-dominant species (*Allan et al., 2015*; *van der Plas et al., 2016*), thus leading to a reduction in individual functions and multifunctionality. In contrast, if functionally dissimilar plant species complement each other in resource uptake and storage strategies, increased resource availability might not reduce species evenness, and instead could increase productivity and support increased and even provisioning of individual functions (*Reich et al., 2001*) and multifunctionality. For example, elevated $CO_2$ concentrations and N addition have been reported to enhance plant diversity effects on plant biomass production (*Reich et al., 2001*) and soil microbial biomass (*Eisenhauer et al., 2013*). Due to these contrasting scenarios and given the importance of soils for ecosystem functioning (*Bardgett and van der Putten, 2014*), evaluating effects of environmental drivers on the multifunctionality of soils and biodiversity-multifunctionality relationships is critically important.

Here, we report root biomass, soil respiration, soil microbial biomass, and water-stable soil aggregates (representing key functions being altered in disturbed soils) (*Amundson et al., 2015*), correlations among these functions, as well as ecosystem multifunctionality (*Hector and Bagchi, 2007*; *Byrnes et al., 2014*) in a long-term grassland plant diversity experiment, with orthogonal manipulation of atmospheric $CO_2$ concentrations and soil N availability in Minnesota, USA (the BioCON experiment) (*Reich et al., 2001*). We measured soil functions in 315 plots, 17 years after establishment of the experimental treatments. To calculate ecosystem multifunctionality, we standardized all functions to values ranging between 0 and 1, and then calculated the average level of ecosystem multifunctionality per plot using the mean of the four standardized functions. To evaluate whether average multifunctionality was a result of plant communities simultaneously performing multiple functions at high levels, we determined ecosystem multifunctionality based on the multiple threshold approach, and we more specifically examined four focal performance thresholds (20, 40, 60, and 80%) (*Byrnes et al., 2014*). The study design enables us to test plant diversity–ecosystem function relationships for multiple soil functions under ambient environmental conditions and under three different global change manipulations that are related to altered resource availability. In addition, we determined treatment effects on realized plant species richness, plant community evenness, the functional diversity of the plant community, and the evenness of multiple soil functions, to explore potential environmental change-induced shifts in the dominance structure, functional diversity, as well as in the even supply of multiple functions (see detailed Materials and methods description below). We find that plant diversity has neutral to positive effects on single soil ecosystem functions and enhances ecosystem multifunctionality. Plant diversity effects on ecosystem multifunctionality are mediated by higher plant community functional diversity and more even provision of multiple soil functions. Although N addition may weaken plant diversity effects on ecosystem multifunctionality at higher thresholds, positive biodiversity-ecosystem functioning relationships were, nevertheless, significant across all the tested global change environments, indicating that conserving local plant diversity is a robust strategy to maintain valuable soil ecosystem services in future environmental conditions.

## Results

### Single-soil ecosystem functions

Plant root biomass, soil respiration, and soil microbial biomass C increased significantly with plant diversity, while the percentage of water-stable soil aggregates increased with plant diversity but only with marginal statistical significance (*Table 1*, *Figure 1*). The identity of top-performing plant species in monoculture depended upon both the response variable and environmental context (Table S1), indicating that different species provide different functions under different conditions. Yet, results for community functioning were independent of the environmental context, as there were no significant interaction effects between plant diversity and global change agents on any individual ecosystem function. In contrast to plant diversity, none of the global change treatments had a significant effect on the single-soil functions (*Table 1*, *Figure 1*).

### Relationships among individual functions

Despite the consistent linear relationships between plant diversity and the soil functions (*Figure 1*), the individual soil functions were generally not, or only weakly correlated with each other across the environmental contexts (in five out of the six possible correlations between soil functions $r^2 \leq 0.1$; *Figure 2*; note that Pearson's correlation coefficients ($r$) are shown in *Figure 2*, while we refer to $r^2$ values in the text). As an exception, a significant proportion of soil respiration was associated with microbial biomass C ($r^2 = 0.45$; *Figure 2*). However, for the positive correlations between root biomass and soil microbial biomass C ($r^2 = 0.10$), and root biomass and respiration ($r^2 = 0.04$), the explained variance did not exceed 10%. Variation in water stable aggregates was not associated with any of the other variables ($r^2 < 0.02$). These mostly weak correlations between soil functions suggest that the single soil functions (as well as ecosystem multifunctionality) may respond largely independently to experimental treatments.

**Table 1.** GLM table of *F* and *p* values on the effects of CO$_2$ (ambient and elevated), N (ambient and elevated), plant species richness (PSR; one, four, nine, or 16 species; log-linear term), and all possible interactions on soil microbial biomass carbon, soil respiration, root biomass, water stable aggregates, and soil ecosystem multifunctionality using the averaging and the multiple thresholds approach.

Ring effects indicate variation across experimental blocks (six rings). Full model: model degrees of freedom (dfs) = 13 (dfs of all factors and interactions = 1, except for Ring [dfs = 6]), error dfs = 301; error dfs for Ring(CO$_2$)=18.56; reduced model without monocultures: error dfs = 177; error dfs for Ring(CO$_2$)=23.65; significant effects (p≤0.05) are given in bold; effects of ring are given in italics.

| | CO$_2$ | | *Ring(CO$_2$)* | | N | | PSR | | CO$_2$ x N | | CO$_2$ x PSR | | N x PSR | | CO$_2$ x N x PSR | |
|---|---|---|---|---|---|---|---|---|---|---|---|---|---|---|---|---|
| | F | p | F | p | F | p | F | p | F | p | F | p | F | p | F | p |
| *Single-soil functions* | | | | | | | | | | | | | | | | |
| Soil microbial biomass | <0.01 | 0.975 | *0.72* | *0.634* | 0.84 | 0.360 | **83.03** | **<0.001** | 0.15 | 0.701 | 0.65 | 0.420 | 0.18 | 0.669 | <0.01 | 0.954 |
| Soil respiration | 0.04 | 0.851 | *5.45* | *<0.001* | 2.91 | 0.089 | **22.98** | **<0.001** | 3.43 | 0.065 | 1.77 | 0.185 | 0.14 | 0.710 | 0.19 | 0.661 |
| Root biomass | 0.25 | 0.621 | *1.77* | *0.105* | 1.15 | 0.285 | **80.23** | **<0.001** | 2.03 | 0.155 | 0.14 | 0.714 | 0.72 | 0.398 | 0.01 | 0.940 |
| Soil aggregate stability | 1.03 | 0.346 | *97.33* | *<0.001* | 0.56 | 0.454 | 2.81 | 0.095 | 0.76 | 0.383 | 0.30 | 0.582 | 0.07 | 0.792 | 0.05 | 0.825 |
| | | | | | | | | | | | | | | | | |
| *Ecosystem multifunctionality (EM)* | 0.58 | 0.457 | *7.53* | *<0.001* | 0.45 | 0.504 | **112.57** | **<0.001** | 3.08 | 0.080 | 0.29 | 0.594 | 0.02 | 0.877 | 0.05 | 0.823 |
| EM, without monocultures' | <0.01 | 0.953 | *8.65* | *<0.001* | 2.90 | 0.091 | **26.87** | **<0.001** | 2.57 | 0.111 | 0.50 | 0.479 | 3.21 | 0.075 | 1.03 | 0.311 |
| EM, realized plant species richness'' | 0.57 | 0.460 | *7.27* | *<0.001* | <0.01 | 0.976 | **91.95** | **<0.001** | 3.75 | 0.054 | 0.07 | 0.788 | 0.01 | 0.904 | 1.13 | 0.290 |
| # Functions > 20% threshold | 0.45 | 0.503 | *1.58* | *0.152* | 1.85 | 0.175 | **79.44** | **<0.001** | 4.34 | 0.038 | 0.80 | 0.373 | 0.86 | 0.355 | 1.46 | 0.222 |
| # Functions > 40% threshold | 0.03 | 0.866 | *1.86* | *0.088* | 0.03 | 0.857 | **124.84** | **<0.001** | 4.24 | 0.041 | 0.22 | 0.636 | 1.57 | 0.211 | 0.02 | 0.876 |
| # Functions > 60% threshold | <0.01 | 0.993 | *4.35* | *0.003* | 0.20 | 0.657 | **64.47** | **<0.001** | 1.18 | 0.278 | 0.03 | 0.867 | 0.02 | 0.889 | 3.87 | 0.050 |
| # Functions > 80% threshold | 0.08 | 0.787 | *21.95* | *<0.001* | 0.43 | 0.514 | **39.21** | **<0.001** | 0.17 | 0.682 | 0.01 | 0.971 | 0.76 | 0.384 | 4.71 | 0.031 |

'without monocultures; only plant species richness levels 4, 9, and 16 used in the analysis

''using realized species richness

DOI: https://doi.org/10.7554/eLife.41228.003

## Ecosystem multifunctionality

Average ecosystem multifunctionality responded strongly to variations in plant diversity, increasing by 40% from monocultures to 16-species mixtures. The strong plant diversity effect on ecosystem multifunctionality was independent of the global change treatments (**Table 1**, **Figure 3**). These results did not depend on the shape of the plant diversity-ecosystem multifunctionality relationship, as additional tests of log-linear, polynomial, and exponential biodiversity-ecosystem multifunctionality relationships yielded qualitatively the same results. Additionally, the plant diversity effect on ecosystem multifunctionality did not depend on the inclusion of monocultures in the statistical analyses (**Table 1**), showing that plant diversity effects on ecosystem multifunctionality were also apparent when diversity increased from four to 16 species. Furthermore, testing the effects of realized, rather than planted, plant species richness did not alter the results (**Table 1**). In contrast to strong plant diversity effects, CO$_2$ and N enrichment or their interactive effects with plant diversity on multifunctionality were not significant (**Table 1**, **Figure 3**).

The multiple threshold approach demonstrates that increases in average ecosystem multifunctionality with increasing plant species richness typically were caused by simultaneous increases in the performance of all functions. However, the extent to which this was true depended upon the choice of threshold criteria for performance and environmental conditions (**Table 1**, **Figure 4**). As the threshold for performance increased from 20 to 80%, plant communities were able to support fewer

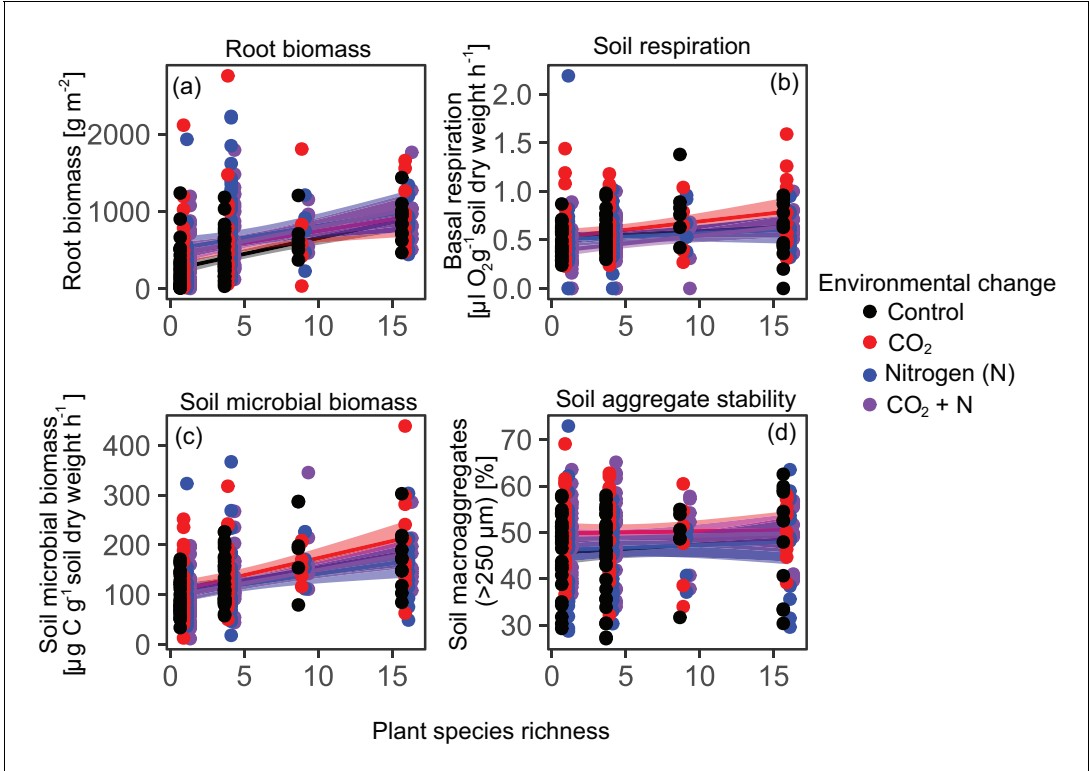

**Figure 1.** Effects of plant diversity and environmental change drivers on single ecosystem functions. Plant diversity effects on root biomass (**a**), soil respiration (**b**), soil microbial biomass (**c**), and water-stable soil aggregates (**d**) under ambient conditions (Control) and three different future environmental conditions (elevated $CO_2$ concentrations, elevated nitrogen availability, and elevated $CO_2$ concentrations and elevated nitrogen availability). Plant diversity effects are significant in (**a**), (**b**), and (**c**), while elevated $CO_2$ and N did not affect any of the soil functions (**Table 1**). Shown are regression lines with 95% confidence intervals. Points staggered on x-axis for clarity.

DOI: https://doi.org/10.7554/eLife.41228.002

functions above threshold values and the influence of diversity on multiple ecosystem functions was more strongly contingent upon environmental conditions (**Figure 4**). At the 20% and 40% thresholds, the interaction between $CO_2$ and N was significant (**Table 1**), with more functions performed above thresholds in ambient conditions and in conditions with both elevated $CO_2$ and elevated N (an effect that was also observed for 60% thresholds), and fewer functions performed above the threshold in conditions of elevated $CO_2$ and elevated N in isolation (**Figure 4**). For the 60% and 80% thresholds, plant species richness effects on the number of functions delivered above the threshold level were strongest under ambient conditions, and the weakest effects were observed under elevated N conditions (significant plant species richness $\times CO_2 \times$ N effect; **Table 1**, **Figure 4**).

## Plant community composition and evenness of multiple soil functions

Sown plant species richness increased realized species richness, the Shannon diversity, evenness, and the functional diversity of the plant community (**Table 2**). Overall, however, this increase in realized diversity with increasing sown plant species richness tended to be less pronounced in the elevated $CO_2$ and N treatments (significant plant species richness $\times$ environmental change effects in **Table 2**, **Figure 4—figure supplement 1**; this pattern was consistent across plant community diversity and functional diversity indices, **Figure 4—figure supplements 2–5**).

Plant species richness increased the Shannon diversity and evenness of multiple soil functions (i.e. the Shannon evenness index of the four standardized soil function values), while $CO_2$ and N as well as interactions among the three factors did not significantly influence these soil responses (**Table 2**). Moreover, structural equation models (SEMs) revealed that the positive effect of plant species richness on ecosystem multifunctionality could be explained by increased plant community functional diversity and evenness of multiple soil functions at high plant diversity (**Figure 5**). These results were

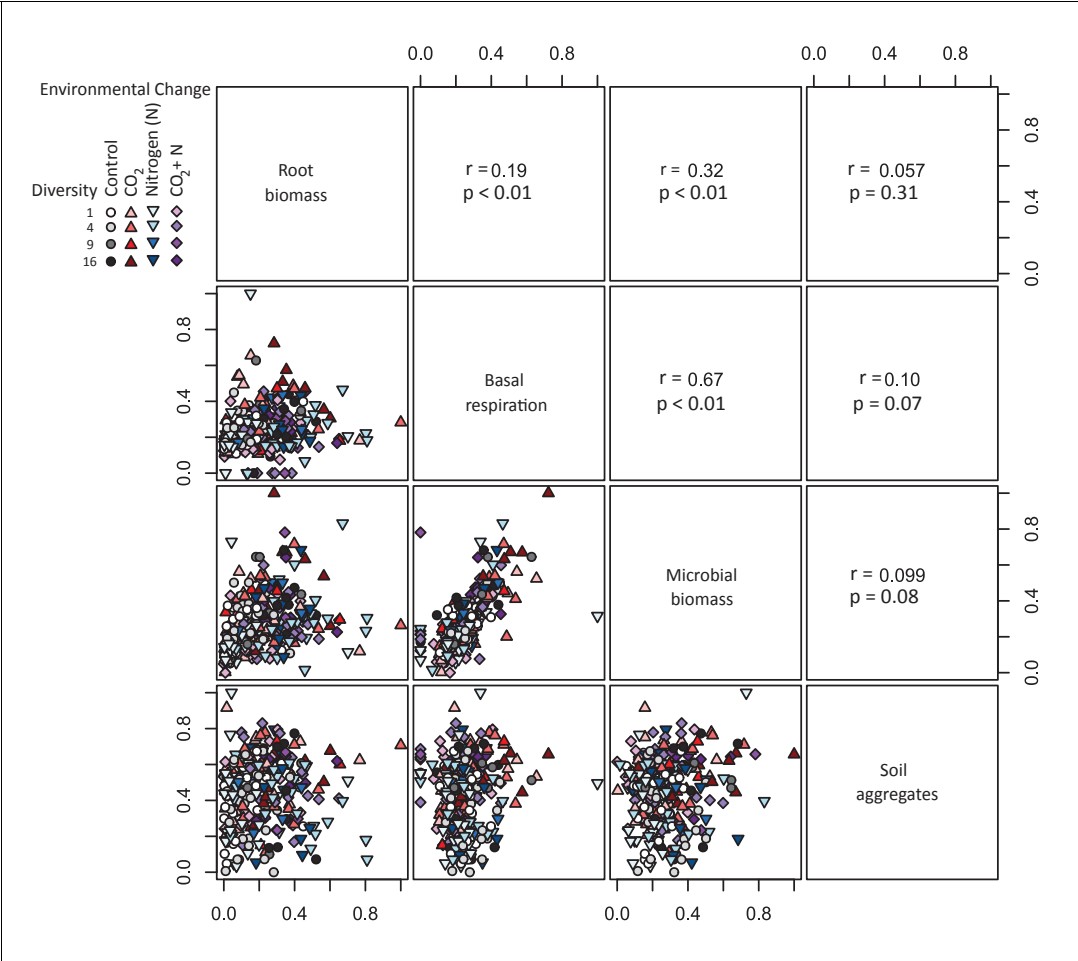

**Figure 2.** Correlations between four soil functions across plant diversity, $CO_2$, and N treatments. Lower panels show bivariate relationships among standardized values of root biomass, soil basal respiration, soil microbial biomass, and water-stable soil aggregates (>250 µm). Upper panels report Pearson's correlation coefficient ($r$) and associated p-values.

DOI: https://doi.org/10.7554/eLife.41228.004

consistent across functional diversity metrics (functional richness, functional evenness, functional diversity, and functional dispersion; not shown), but we focus on functional dispersion here, because this diversity index could be calculated for all 315 plots, while the other functional diversity indices need a minimum number of three species present in order to be calculated. No direct paths between plant species richness and ecosystem multifunctionality were supported by the SEM, indicating that plant species richness effects on ecosystem multifunctionality were fully explained by changes in the functional diversity of the plant community and evenness of ecosystem multifunctionality. Although several significant interaction effects for plant species richness × $CO_2$ (Shannon diversity, Shannon evenness, functional dispersion) and plant species richness × N (realized species richness, Shannon diversity, Shannon evenness, Simpson evenness, functional dispersion) on the plant community were observed (*Table 2*), $CO_2$ and N were not retained in the final SEM (removal of non-significant paths improved the model fit based on AIC). Taken together, these results suggest that strong plant species richness effects overrode any $CO_2$ and N effects on plant community evenness, functional diversity, and evenness of multiple soil functions in driving ecosystem multifunctionality (*Table 1*, *Figure 5*).

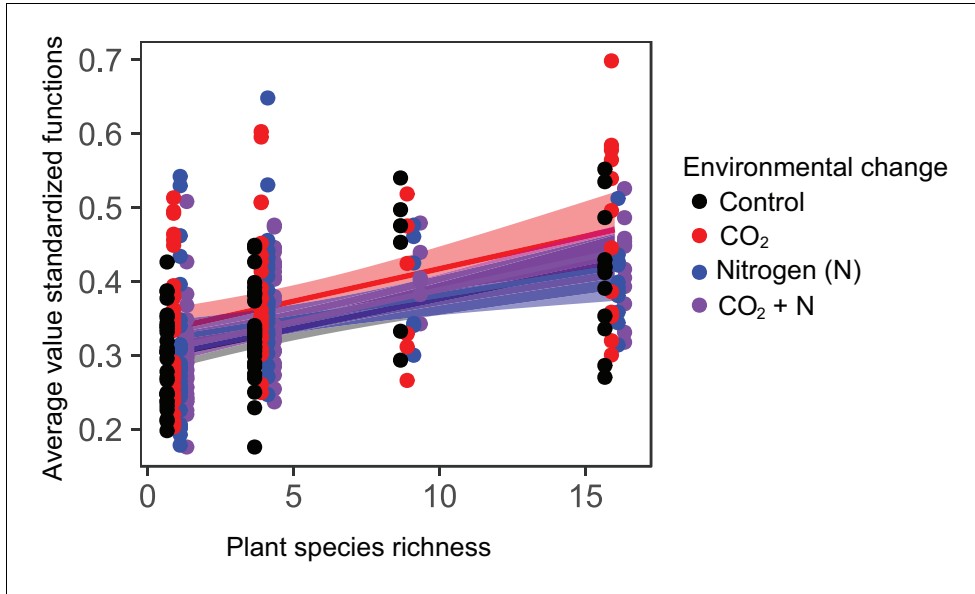

**Figure 3.** Average ecosystem multifunctionality as affected by plant diversity and environmental change drivers. Average ecosystem multifunctionality (*Hector and Bagchi, 2007*; *Byrnes et al., 2014*) calculated from the soil functions root biomass, soil respiration, soil microbial biomass C, and percentage of water-stable soil aggregates under ambient conditions and three different future environmental conditions (elevated $CO_2$ concentrations, elevated nitrogen availability, and elevated $CO_2$ concentrations and elevated nitrogen availability). Plant diversity significantly increased multifunctionality, while $CO_2$ and N effects were not significant (*Table 1*). Given are regression lines with 95% confidence intervals. Points staggered on x-axis for clarity.
DOI: https://doi.org/10.7554/eLife.41228.005

## Discussion

While it is well established that increased plant diversity (*Hector et al., 1999*) and resource availability (*Lee et al., 2010*) enhance the functioning of grasslands, potential interactive effects of these factors are less well studied and are mostly based on responses of single functions like primary productivity (*Isbell et al., 2015*; *Craven et al., 2016*; *Reich et al., 2001*; *Stocker et al., 1999*). Our study shows the first empirical evidence that significant plant diversity effects on multiple soil functions, and therefore ecosystem multifunctionality, are largely robust to changes in environmental conditions caused by elevated atmospheric $CO_2$ concentrations, N inputs, and both factors in combination. That is, although there was turnover in the identity of best performing monocultures for each response variable and each global change scenario, the biodiversity effects held in three broad contexts that may occur in the future (*Craven et al., 2016*; *Loreau and Hector, 2001*). However, plant diversity effects on ecosystem multifunctionality at higher thresholds of functionality may be impaired under future environmental conditions (*Figure 4*), potentially by reducing the taxonomic and functional diversity of the plant community (*Figure 5*). Plant diversity effects on ecosystem multifunctionality were mediated by higher levels of plant community functional diversity and the more even provisioning of multiple soil functions. Our measurements of root biomass, soil respiration, soil microbial biomass C, and percentage of water-stable soil aggregates are indicators of belowground plant biomass production, organic matter decomposition, soil carbon storage (*Lange et al., 2015*), and soil erosion control, respectively, which denote some of the most crucial ecosystem services grasslands provide for human well-being (*Millennium Ecosystem Assessment, 2005*). Thus, management policies targeted to maximize these services through maintaining or increasing plant diversity should be applicable under a wide range of contexts (*Manning et al., 2018*), although high availability of certain resources may compromise beneficial plant diversity effects on high levels of functioning.

Previous studies, including from this field experiment (*Reich, 2009*), have shown that the composition, species richness, and functioning (e.g. plant biomass production) of grasslands can be

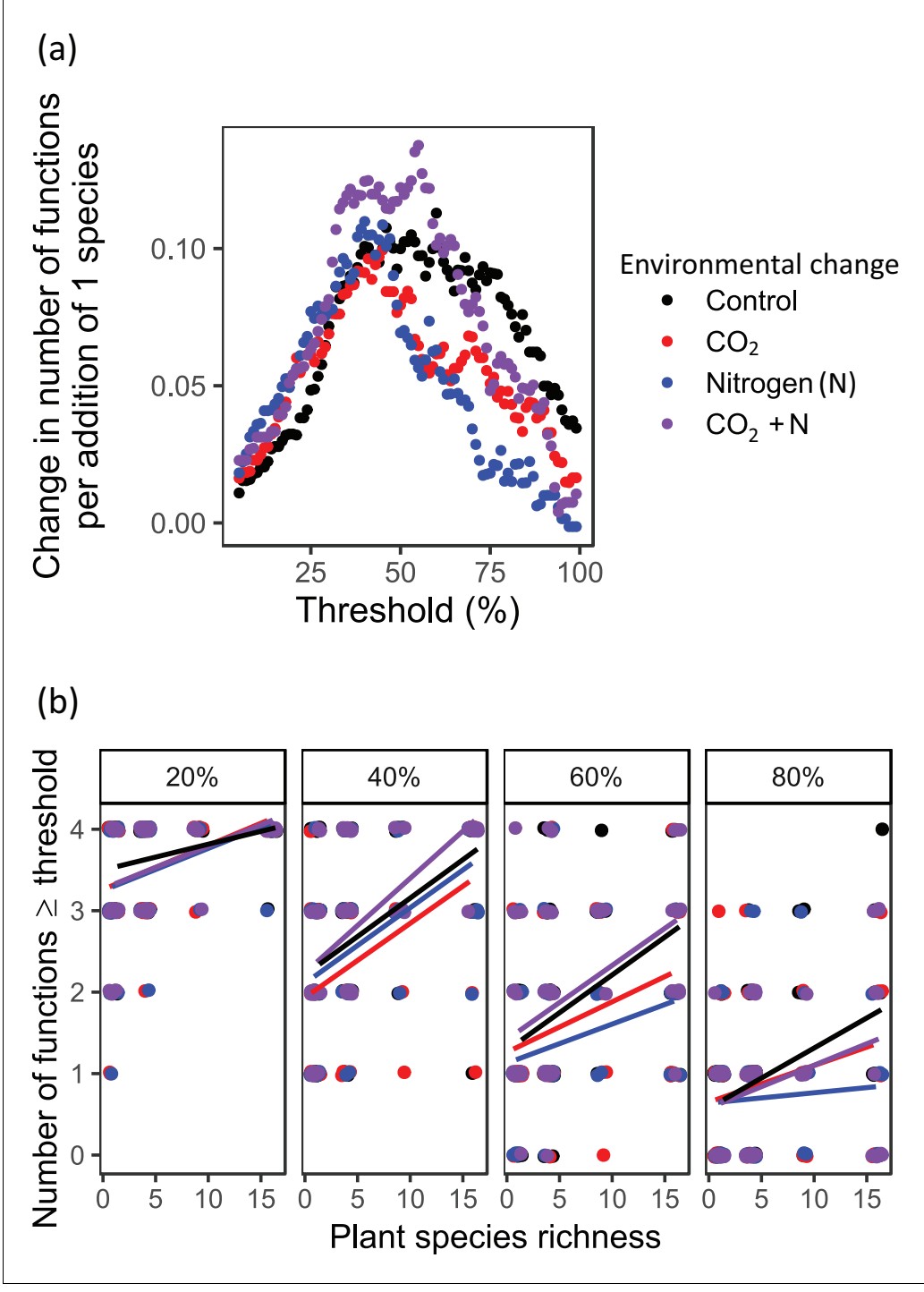

**Figure 4.** Ecosystem multifunctionality based on the multiple thresholds approach (*Byrnes et al., 2014*) as affected by plant diversity and environmental change drivers. The slope of the relationship between planted species richness and multifunctionality, defined as number of functions reaching a threshold of some percentage of the maximum observed function. Panels show the relationship for (**a**) the number of functions at or above a threshold of some proportion of the maximum observed function for threshold values ranging from 1 to 99% and (**b**) four different thresholds (20%, 40%, 60%, and 80% of maximum) as affected by plant species richness, $CO_2$, and N in the BioCON experiment (*Reich et al., 2001*). Points are slightly jiggered to improve readability.
DOI: https://doi.org/10.7554/eLife.41228.006

The following figure supplements are available for figure 4:

*Figure 4 continued on next page*

*Figure 4 continued*

**Figure supplement 1.** Plant species richness, N addition, and $CO_2$ effects on *realized species richness* in the BioCON experiment.
DOI: https://doi.org/10.7554/eLife.41228.007
**Figure supplement 2.** Plant species richness, N addition, and $CO_2$ effects on *Shannon diversity* in the BioCON experiment.
DOI: https://doi.org/10.7554/eLife.41228.008
**Figure supplement 3.** Plant species richness and N addition effects on *Simpson evenness* in the BioCON experiment.
DOI: https://doi.org/10.7554/eLife.41228.009
**Figure supplement 4.** Plant species richness and $CO_2$ effects on *Functional Dispersion* in the BioCON experiment.
DOI: https://doi.org/10.7554/eLife.41228.010
**Figure supplement 5.** Plant species richness and N addition effects on *Functional Dispersion* in the BioCON experiment.
DOI: https://doi.org/10.7554/eLife.41228.011

significantly affected by anthropogenic environmental changes, such as climate change (*Reich et al., 2014b*; *Grime et al., 2000*), elevated atmospheric $CO_2$ concentrations (*Reich et al., 2001*;

**Table 2.** GLM table of *F* and *p* values on the effects of $CO_2$ (ambient and elevated), N (ambient and elevated), plant species richness (PSR; one, four, nine, or 16 species; log-linear term), and all possible interactions on realized plant species richness, Shannon diversity index of plants, Simpson evenness of plants, aboveground plant biomass, Shannon diversity index of soil functions, and evenness of soil functions.
Ring effects indicate variation across experimental blocks (six rings). Model degrees of freedom (dfs) = 13 (dfs of all factors and interactions = 1, except for Ring [dfs = 6]), error dfs = 301; error dfs for Ring($CO_2$)=18.56; significant effects (p<0.05) are given in bold; effects of ring are given in italics.

| | $CO_2$ | | *Ring(CO$_2$)* | | N | | PSR | | $CO_2$ x N | | $CO_2$ x PSR | | N x PSR | | $CO_2$ x N x PSR | |
|---|---|---|---|---|---|---|---|---|---|---|---|---|---|---|---|---|
| | *F* | *p* | *F* | *p* | *F* | *p* | *F* | *p* | *F* | *p* | *F* | *p* | *F* | *p* | *F* | *p* |
| *Plant community responses* | | | | | | | | | | | | | | | | |
| Realized plant species richness | 0.01 | 0.938 | *1.11* | *0.355* | <0.01 | 0.949 | **1040.30** | **<0.001** | 0.64 | 0.424 | 0.01 | 0.908 | **12.34** | **<0.001** | **7.20** | **0.008** |
| Shannon diversity | 0.82 | 0.365 | *0.86* | *0.526* | 0.40 | 0.528 | **761.67** | **<0.001** | 0.26 | 0.590 | **5.81** | **0.017** | **11.62** | **<0.001** | **5.03** | **0.026** |
| Shannon evenness | 0.11 | 0.736 | *0.40* | *0.880* | 0.27 | 0.604 | **445.43** | **<0.001** | <0.01 | 0.982 | 0.12 | 0.726 | **5.02** | **0.026** | 0.09 | 0.965 |
| Simpson evenness | 0.91 | 0.342 | *1.05* | *0.392* | 0.72 | 0.397 | **622.34** | **<0.001** | 0.09 | 0.763 | **6.28** | **0.013** | **8.05** | **0.005** | 3.43 | 0.065 |
| Functional richness | 0.06 | 0.802 | *0.44* | *0.853* | 0.03 | 0.874 | **597.98** | **<0.001** | **0.04** | 0.842 | 0.19 | 0.667 | 0.13 | 0.724 | 1.41 | 0.236 |
| Functional evenness | 1.68 | 0.203 | *3.33* | *0.007* | 1.16 | 0.283 | **30.52** | **<0.001** | 2.94 | 0.089 | 2.36 | 0.127 | 0.29 | 0.592 | 0.87 | 0.353 |
| Functional divergence | 0.46 | 0.498 | *0.96* | *0.455* | 0.61 | 0.434 | **970.50** | **<0.001** | 0.11 | 0.740 | 0.67 | 0.415 | 0.15 | 0.703 | 0.51 | 0.475 |
| Functional dispersion | 0.57 | 0.453 | *1.12* | *0.353* | 0.35 | 0.555 | **905.91** | **<0.001** | 0.02 | 0.893 | **7.71** | **0.006** | **4.71** | **0.031** | 0.71 | 0.399 |
| Aboveground plant biomass | 0.99 | 0.323 | *1.52* | *0.171* | **6.71** | **0.010** | **65.37** | **<0.001** | 0.54 | 0.463 | 0.28 | 0.596 | 0.01 | 0.926 | 0.20 | 0.657 |
| *Ecosystem multifunctionality* | | | | | | | | | | | | | | | | |
| Shannon diversity index of soil functions | 0.05 | 0.827 | *5.93* | *<0.001* | 2.03 | 0.155 | **115.34** | **<0.001** | 0.91 | 0.340 | 0.43 | 0.512 | 1.39 | 0.234 | 0.13 | 0.718 |
| Evenness of soil functions | 0.11 | 0.652 | *6.24* | *<0.001* | 0.58 | 0.449 | **135.63** | **<0.001** | 0.80 | 0.371 | 0.46 | 0.500 | 0.96 | 0.327 | 0.51 | 0.477 |

DOI: https://doi.org/10.7554/eLife.41228.012

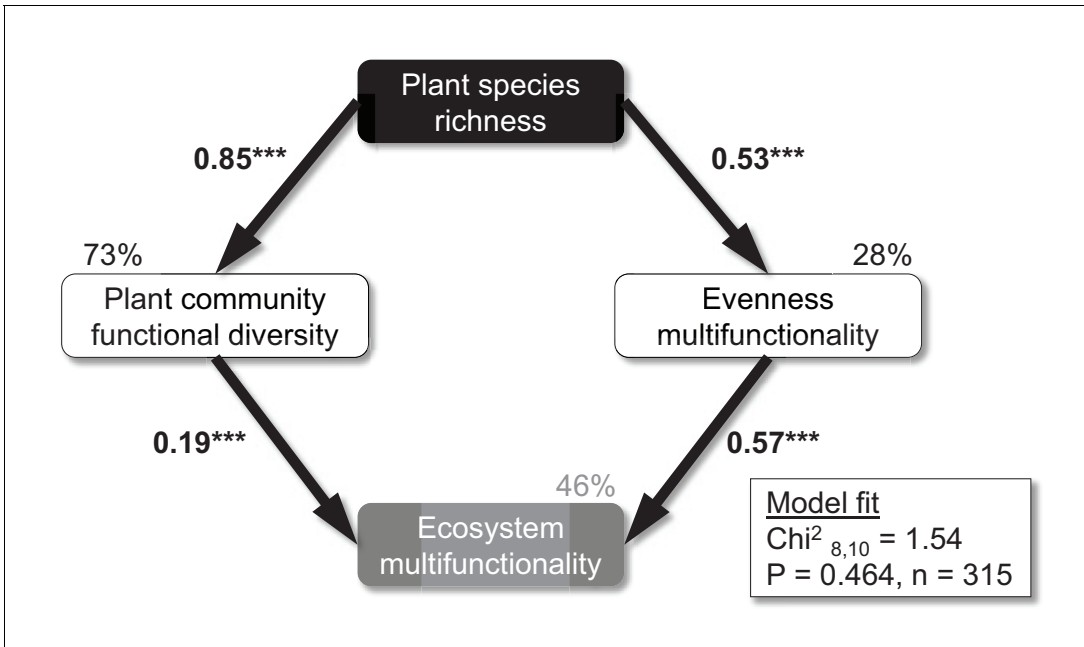

**Figure 5.** Mechanisms underlying plant diversity effects on ecosystem multifunctionality. Structural equation model on the effects of sown plant species richness on the functional diversity of the plant community (functional dispersion), evenness of multiple functions, and ecosystem multifunctionality. Numbers on arrows are standardized path coefficients. All paths retained in this final model were significant (p<0.001). Percentages on boxes indicate the variance explained by the model. Modification indices were used to check if additional paths would improve the model. ***p<0.001.
DOI: https://doi.org/10.7554/eLife.41228.013

*Morgan et al., 2011*), and N deposition (*Clark and Tilman, 2008*; *Simkin et al., 2016*). In contrast, in the present study, elevated $CO_2$ concentrations and N input had no significant effects on soil functions (*Table 1*), despite significantly increased aboveground plant biomass in the N addition treatment (*Table 2*). These overall weak or missing effects might be because the response of grassland functions to environmental change depends on local conditions like precedent climate events and history of anthropogenic disturbances (*Grime et al., 2000*; *Simkin et al., 2016*), although we did not record any particularly harsh environmental conditions in the year of the study. Previous research in the BioCON experiment showed significant $CO_2$ and N effects on plant biomass (*Reich et al., 2001*) and total belowground carbon allocation, as well as on soil communities (*He et al., 2010*; *Eisenhauer et al., 2012b*) and functions (*Chung et al., 2007*; *Reich et al., 2018*). However, the effect of those global change drivers on total belowground carbon allocation and soil functions may increase or decrease over time and vary among years and species compositions alone or jointly (*Reich et al., 2018*; *Adair et al., 2009*; *Andresen et al., 2016*). The long-term character of the BioCON experiment (17 y of experimental treatments in the sampling year) (*Reich et al., 2001*), the gradual increase of biodiversity effects over time (*Reich et al., 2012*), and the fact that some soil functions, such as root biomass and soil aggregate stability, should reflect treatment effects of multiple years, suggest that the present results are unlikely to be due to the fact that they are based on a single sampling campaign. This is supported by previous assessments of soil respiration (*Eisenhauer et al., 2013*), soil microbial biomass (*Eisenhauer et al., 2013*), and root biomass (*Reich et al., 2001*) that showed significant positive plant diversity effects. However, environmental change effects may be more important if higher thresholds of functionality are considered (*Figure 4*), showing that the multiple thresholds approach can provide additional insights into the functioning of ecosystems.

In contrast to the effects of the other two global change drivers, our study showed that plant diversity had strong and consistent effects on three out of the four single soil functions we investigated, as well as on ecosystem multifunctionality. In line with other studies (*Reich et al., 2012*; *Mueller et al., 2013*; *Ravenek et al., 2014*), we observed positive effects of plant diversity on root biomass. High-diversity plant communities are more likely to include different root traits, that is

deep- and shallow-rooting species, than low-diversity communities (*Mueller et al., 2013*). This may lead to a more complete usage of the available habitat space, thus increasing total resource extraction from the soil (*Ravenek et al., 2014*; *Mommer et al., 2010*). Thereby, higher diversity plant communities have been shown to increase root biomass in surface soils (*Ravenek et al., 2014*) and the proportion of deep roots (*Mueller et al., 2013*). Increased plant biomass production at high plant diversity (above and below the ground) and increased diversity of organic inputs to the soil have been shown to enhance soil microbial biomass and activity (*Hooper et al., 2000*; *Eisenhauer et al., 2013*; *Lange et al., 2015*). This was also found in the present study (*Tables 1* and *2*, *Figure 1b,c*) and indicates higher decomposition activity and turnover of microbial products with potential consequences for soil carbon storage (*Lange et al., 2015*). The simultaneous increase in functions related to soil carbon storage (e.g. soil microbial biomass and root biomass) and decomposition activity (soil respiration) may appear counterintuitive as one may expect trade-offs between these functions. However, these results are in line with recent findings showing that elevated microbial activity, such as indicated by soil respiration, reflects the transfer of higher amounts of plant products into microbial products that then accumulate in the soil (*Lange et al., 2015*; *Cotrufo et al., 2015*). Thus, although high soil respiration means greater short-term losses of organic matter as $CO_2$, it can also indicate enhanced long-term soil carbon storage (*Lange et al., 2015*; *Cotrufo et al., 2015*) as microbial necromass might end up in slow-cycling soil organic matter pools (*Cotrufo et al., 2015*; *Schmidt et al., 2011*).

In contrast to previous studies, the proportion of water-stable soil aggregates did not significantly increase with plant diversity (*Gould et al., 2016*; *Pérès et al., 2013*), root biomass (*Gould et al., 2016*), microbial biomass (*Gould et al., 2016*; *Pérès et al., 2013*), elevated $CO_2$ (*Rillig et al., 2001*), or N addition (*Riggs et al., 2015*), although positive trends were observed with diversity (*Figure 1d*). Soil aggregate stability contributes to the regulating ecosystem service soil erosion control (*Millennium Ecosystem Assessment, 2005*), which is of major relevance in a rapidly changing world with increased likelihood of extreme weather events (*Amundson et al., 2015*; *Fischer et al., 2018*). Soil aggregate stability also informs soil carbon storage, as aggregates incorporate organic matter protecting it from decomposition (*Six et al., 2000*) on decadal timescales. Although soil aggregate stability depends on soil biological properties, such as (fine) root biomass and root length density (*Gould et al., 2016*), and soil biota including microbes and animals (*Lehmann et al., 2017*), abiotic factors like soil texture are also important controls (*Denef et al., 2002*) and may have prevailed in the very sandy soil of the present study (*Reich et al., 2001*).

Our study also showed the importance of plant diversity for the simultaneous provisioning of multiple functions (soil multifunctionality). Despite the increasing awareness of the importance of biodiversity for ecosystem multifunctionality (*Lefcheck et al., 2015*; *Manning et al., 2018*; *Hector and Bagchi, 2007*; *Soliveres et al., 2016*; *Byrnes et al., 2014*), knowledge of how these results hold under future environmental conditions is limited. Increases in N deposition and in $CO_2$ levels are predicted with future global change and represent elevated resource availability. Elevated resources can facilitate the dominance of certain species that are more efficient in resource use leading to reduced taxonomic and functional diversity of plant communities (*Morgan et al., 2011*; *Clark and Tilman, 2008*; *Simkin et al., 2016*). This, in turn, could be expected to offset the observed positive effects of plant diversity on multiple ecosystem functions (*Hector and Bagchi, 2007*; *Soliveres et al., 2016*; *Byrnes et al., 2014*). Indeed, elevated resource availability decreased realized plant species richness, community evenness, and functional diversity in the present study (*Table 2*, *Figure 4—figure supplements 1–5*). However, our results indicate that higher sown plant diversity attenuated the detrimental effects of elevated resource availability on realized plant diversity indices, and we found that positive effects of plant diversity on single ecosystem functions, as well as on soil multifunctionality, were largely independent of environmental conditions. Although these results were robust to different analytical approaches to quantify multifunctionality (*Byrnes et al., 2014*), and also hold in realistic scenarios where plant diversity loss is relatively moderate (*Eisenhauer et al., 2013*) (excluding monocultures), plant diversity effects on ecosystem multifunctionality were attenuated for high thresholds (*Byrnes et al., 2014*) under changed environmental conditions. Thus, our study provides strong evidence that the positive effects of plant diversity on soil functioning are not only applicable at present, but also in realistic future global change scenarios. However, our results also indicate that environmental conditions decreasing the functional diversity of plant communities may threaten the multifunctionality of soils.

The present and other grassland studies have shown that biodiversity effects on aboveground functions (aboveground productivity) can also be robust to changes in N addition and water availability (*Craven et al., 2016*). This contrasts with other findings on context-dependent biodiversity effects (*Guerrero-Ramírez et al., 2017*; *Ratcliffe et al., 2017*). Future studies should investigate the causes of these inconsistent findings, for example by considering variations in the experimental age (*Craven et al., 2016*; *Guerrero-Ramírez et al., 2017*; *Thakur et al., 2015*), local biotic and abiotic conditions (*Guerrero-Ramírez et al., 2017*; *Ratcliffe et al., 2017*), and the focal functions (*Byrnes et al., 2014*; *Allan et al., 2015*). While the effects of biodiversity on certain ecosystem functions in certain habitat types may become stronger or weaker under future conditions, we show that grassland biodiversity effects on at least the belowground functions we studied are robust to environmental change as long as high levels of plant functional diversity are maintained. Thus, studying shifts in the functional composition of plant communities in response to environmental change is a promising approach to better understand the context dependency of biodiversity effects based on species richness.

In summary, our study showed strong and consistent effects of plant diversity on various belowground ecosystem functions related to biomass production, nutrient cycling, and carbon sequestration, as well as on ecosystem multifunctionality. These effects were much stronger than those of two other global change drivers, stressing the pivotal importance of plant diversity (*Hooper et al., 2012*; *Thakur et al., 2015*) for soil functioning. The present study sheds light on the underlying mechanisms of plant diversity effects on ecosystem multifunctionality by stressing the role of plant community functional diversity as well as the more even provisioning of multiple ecosystem functions at high plant diversity. Importantly, we also show that these positive effects of biodiversity were largely robust to environmental changes in N deposition and $CO_2$ levels, indicating that biodiversity will be equally important under future conditions as at present. Notably, high levels of soil multifunctionality may be threatened by environmental change-induced reductions in the functional diversity of plant communities.

## Materials and methods

### Experimental setup

This experiment was conducted in the framework of the BioCON experiment at the Cedar Creek Long Term Ecological Research (LTER) site in Minnesota (*Reich et al., 2001*). The region has a continental climate with cold winters (mean January temperature −11°C) and warm summers (mean July temperature 22°C) and mean annual precipitation of 660 mm (*Reich et al., 2001*). The soils are sands (Typic Udipsamment, Nymore series) derived from sandy glacial outwash (94.4% sand, 2.5% clay). The BioCON experiment was designed for the simultaneous manipulation of plant diversity (1, 4, 9, 16 species), atmospheric $CO_2$ concentrations (ambient, elevated), and N deposition (ambient, elevated) in experimental grassland plots (2 × 2 m) under field conditions, using a well-replicated split-plot experiment comprising a full-factorial combination of treatment levels (orthogonal cross of all plant diversity × $CO_2$×N treatments) in a completely randomized design (*Reich et al., 2001*). It was established in 1997 on a level, secondary successional grassland after removing prior vegetation (*Reich et al., 2001*), and experimental treatments had been continuously ongoing since 17 years before the present study was conducted in the summer of 2014. The established plant diversity levels represent common plant species richness numbers per square meter in the study region and cover the range from disturbed grassland of anthropogenic origin to medium-high diversity native vegetation (*Eisenhauer et al., 2013*).

Plots within each of six circular areas (half at ambient, half at elevated $CO_2$) are separated by a 20 cm walkway buffer, and metal barriers 30 cm deep separate each plot. Plots were planted in 1997 from a pool of 16 herbaceous plant species from four functional groups (C3 grasses, C4 grasses, legumes, and non-leguminous forbs). All species were planted in replicate monocultures at each $CO_2$ x N level. In total, there were 32, 32, 15, and 12 replicates at each $CO_2$ x N level for 1, 4, 9, and 16 species plots, respectively (91 replicates x 4 $CO_2$/N treatments = 364 plots in total; as some measurements did not work for single samples, data from 315 to 349 plots entered the present analyses). The species pool comprised the C3 grasses *Agropyron repens*, *Bromus inermis*, *Koeleria cristata*, and *Poa pratensis*; the C4 grasses *Andropogon gerardii*, *Bouteloua gracilis*, *Schizachyrium*

*scoparium*, and *Sorghastrum nutans*; the herbaceous forbs *Achillea millefolium*, *Anemone cylindrica*, *Asclepias tuberosa*, *Solidago rigida*; and the N-fixing legumes *Amorpha canescens*, *Lespedeza capitata*, *Lupinus perennis*, and *Petalostemum villosum*. Four species-mixtures contained either 1, 2, 3, or 4 plant functional groups, thus representing the whole gradient of plant functional group levels within the same species richness level. Nine plant species-mixtures almost all contained four plant functional groups, a few had three functional groups, while sixteen plant species-mixtures always contained all four plant functional groups.

$CO_2$ treatments consist of ambient and elevated $CO_2$. Six circular areas (24 m diameter) were randomly assigned, three each to ambient and elevated $CO_2$ (+180 ppm, during daylight hours from early spring to late fall). The added $CO_2$ is delivered using FACE technology (*Reich et al., 2001*). Such a rise in atmospheric $CO_2$ concentrations may be expected in the 21st century according to a broad range of $CO_2$ emission scenarios (*Meinshausen et al., 2009*).

Nitrogen was added to the surface of half the plots in each ring as 4 g N $m^{-2}$ $yr^{-1}$ slow-release ammonium nitrate ($NH_4NO_3$) in equal fractions in early May, June, and July. Annual net mineralization rates are roughly 3–4 g N $m^{-2}$ $y^{-1}$ in grassland at Cedar Creek. Thus, adding 4 g N $m^{-2}$ $y^{-1}$ doubles available N in this system and serves to elucidate responses of ecosystems differing in soil N supply because of differences in fertility or N deposition (*Eisenhauer et al., 2013*). The levels of experimentally increased N deposition rates are already realized across various locations around the globe (*Simkin et al., 2016*).

## Samplings and measurements

In August 2014 (during the period of peak plant biomass), we determined plant aboveground biomass (in an area of 0.1 $m^2$ per plot), realized plant species richness, Shannon diversity and evenness, as well as functional richness, functional divergence, functional evenness, and functional dispersion (*Villéger et al., 2008*; *Laliberté and Legendre, 2010*) of the plant community at the plot level. Realized species richness, Shannon diversity and evenness, Simpson diversity and functional diversity indices of the communities were calculated based on plot- and species-specific cover estimates. We assembled trait data of all plant species used in this analysis representing a wide range of the global spectrum of plant forms and functions (*Díaz et al., 2016*): plant height (H), specific leaf area (SLA), leaf dry matter content (LDMC), leaf nitrogen concentration (leafN), and seed mass. Those traits reflect the capacity of light preemption (H), resource capturing (SLA, LDMC, leafN), and reproduction (seed mass). We derived data on seed mass and leafN from monocultures of the BioCON experiment (*Reich et al., 2001*), considering site- and treatment-specific variability in the traits. SLA and H were derived from monocultures of a nearby experiment; thus, these datasets do not reflect treatment-specific variability. LDMC data were derived from the TRY database (*Kattge et al., 2011*). Functional richness, functional evenness, functional divergence (*Villéger et al., 2008*), and functional dispersion (*Laliberté and Legendre, 2010*) were calculated with the function dbFD of the package FD in R 3.3.3. We weighted those indices by the relative abundances (plot cover) of the present target plant species. In cases where all sown species of a community went extinct, we set all diversity indices to zero.

In June and August 2014, we took soil samples to investigate treatment effects on multiple soil functions. From each of the 364 plots, we took four 2 cm diameter soil samples (to 20 cm depth) for soil microbial and soil aggregate stability analyses (June), and three 5 cm diameter soil samples (to 20 cm depth) for determination of root biomass (August) using steel corers. The soil samples were pooled in plastic bags (separately for microbial and root analyses), carefully but thoroughly homogenized, and stored at 4°C until further processing. Investigating plant diversity effects in four $CO_2$ x N scenarios allowed us to study effects in different environmental contexts (one ambient and three global change scenarios).

Roots were washed, dried, and weighed (g $m^{-2}$). Before measurement of soil microbial parameters, soil sub-samples were sieved (2 mm) to remove roots (*Eisenhauer et al., 2013*). Soil microbial biomass C and respiration of approximately 5 g soil (fresh weight) was measured using an $O_2$-microcompensation apparatus (*Scheu, 1992*). The microbial respiratory response was measured at hourly intervals for 24 hr at 20°C. Soil respiration (µl $O_2$ $h^{-1}$ $g^{-1}$ soil dry weight) was determined without addition of substrate and measured as mean of the $O_2$ consumption rates of hours 14 to 24 after the start of the measurements. Substrate-induced respiration was calculated from the respiratory response to D-glucose for 10 hr at 20°C (*Eisenhauer et al., 2013*). Glucose was added according to

preliminary studies to saturate the catabolic enzymes of microorganisms (4 mg g$^{-1}$ dry weight dissolved in 400 µl deionized water). The mean of the lowest three readings within the first 10 hr (between the initial peak caused by disturbing the soil and the peak caused by microbial growth) was taken as maximum initial respiratory response (MIRR; µl O$_2$ g$^{-1}$ soil dry weight h$^{-1}$) and microbial biomass (µg C g$^{-1}$ soil dry weight) was calculated as 38 × MIRR (*Beck et al., 1997*).

To determine the resistance of soil aggregates against water as a disintegrating force, we applied an approach modified from Kemper and Rosenau (*Kemper and Rosenau, 1986*). The resulting index represents the percentage of water-stable aggregates with a diameter smaller than 4 mm. Dry soil (4.0 g, measured in duplicates) was placed onto small sieves with a mesh size of 250 µm, capillarily re-wetted with deionized water prior, and then placed in a sieving machine (Agrisearch Equipment, Eijkelkamp, Giesbeek, Netherlands) where the samples were agitated for 3 min. The re-wetting and agitation of the tested soil aggregates causes the compression of entrapped air inside of them resulting in a process called slaking, which is a function of re-wetting intensity, volume of entrapped air, and aggregate shear-strength (*Bissonnais, 1996*). This process leads to a separation into water-stable and water-unstable fraction with a size >250 µm. Additionally, debris (i.e. coarse matter) had to be separated from the water-stable fraction to correctly determine the water-stable aggregates (WSA) fraction of the sample:

$$\%WSA = (water\ stable\ fraction - coarse\ matter)/(4\ g - coarse\ matter).$$

The focal soil functions were carefully chosen to (i) represent different soil-related ecosystem services and (ii) to not be too tightly correlated with each other. Plant root biomass is an indicator of belowground primary production and is often related to soil carbon storage (*Lange et al., 2015*) and soil erosion control (*Gyssels et al., 2005*). While soil respiration indicates microbial decomposition activity (*Lange et al., 2015*), soil microbial biomass is a proxy for belowground secondary production, soil enzyme and phosphorous dynamics (*Hacker et al., 2015*), soil nitrogen leaching (*Leimer et al., 2016*), and both variables are powerful predictors of soil carbon storage (*Thakur et al., 2015*; *Lange et al., 2015*) and the natural attenuation of polycyclic aromatic compounds (*Bandowe et al., 2018*). The percentage of water-stable soil aggregates indicates soil stability and may be an important determinant of soil erosion control and soil sustainability (*Lehmann et al., 2017*). Given the tight correlation between soil carbon concentrations and soil microbial biomass C (*Lange et al., 2015*; *Eisenhauer et al., 2010*), and because soil microbial biomass has been shown to be a significant predictor of many soil functions (see above), we focused on the latter in the present study.

## Calculations and statistical analyses

We report the top performing monoculture in each environment/function to explore if the same or different species performed well under different environmental conditions (*Supplementary file 1*). We assessed ecosystem multifunctionality with the averaging and the multiple thresholds approach (*Byrnes et al., 2014*). Briefly, to calculate average ecosystem multifunctionality, we standardized all functions to values ranging between 0 and 1, and then calculated the average level of ecosystem multifunctionality per plot as the mean of the four standardized functions. We are aware of the advantages and disadvantages of presenting aggregate measures of ecosystem multifunctionality (for review see *Manning et al., 2018*), which is why we put equal emphasis on the results based on the four single focal soil functions and show results of the multiple thresholds approach (*Figure 4*). To evaluate whether multiple functions are simultaneously performing at high levels, we created an index of the number of functions surpassing different thresholds in each experimental plot. This threshold reflects the percentage of the maximum observed value of each function (R package 'multifunc'; *Byrnes et al., 2014*). Although we present the slope of the relationship between planted species richness and the number of functions at or above a threshold of some proportion of the maximum observed function for threshold values ranging from 1 to 99% (*Figure 4a*), we focused on the number of functions for four different thresholds (20%, 40%, 60%, and 80% of maximum; *Figure 4b*) of percentage of the maximum observed function for statistical analyses (*Byrnes et al., 2014*). Moreover, we determined the evenness of multiple soil functions by treating the four functions like different species and calculating evenness based on the standardized values of these functions per plot.

General linear models (GLMs, type III sum of squares) were used to measure the effects of plant species richness, CO$_2$, N, and all interactions (*Supplementary file 2*) on the four single soil functions,

ecosystem multifunctionality (based on the averaging approach and the multiple thresholds approach for 20%, 40%, 60%, and 80%), realized plant species richness, Shannon diversity, plant community evenness and functional diversity, as well as evenness of multiple soil functions (*Tables 1* and *2*). The effect of $CO_2$ was tested against the random effect of ring nested within $CO_2$ (*Supplementary file 2*). This creates a caveat regarding the comparison of $CO_2$ with species richness, because size of effects may differ due to different levels of replication. Moreover, we were not able to test for interactive effects of plant species richness and environmental change drivers against species composition effects, because species composition changed significantly over time in response to the experimental treatments (*Table 2*; *Schmid et al., 2017*). In addition, we performed sensitivity analyses by running the same statistical models for ecosystem multifunctionality without the inclusion of plant monocultures to test if plant diversity effects are solely due to that plant diversity level (*Table 1*), and we explored treatment effects on realized plant community Shannon diversity and evenness as well as the functional diversity of the plant community (*Table 2*). Moreover, we used the same statistical model with realized plant species richness on the study plots to account for potential effects of $CO_2$ and N through changes in plant diversity. These analyses were complemented by structural equation modeling (SEM) (*Grace, 2006*) to explore potential direct and indirect (through plant community functional diversity and evenness of multiple soil functions) global change effects on ecosystem multifunctionality in a multivariate analysis. The model fit was determined via $\chi^2$ tests, and the initial model with all hypothesized paths was modified to achieve a good model fit (*Figure 5*): removal of direct paths from plant species richness to ecosystem multifunctionality and the correlation between functional diversity of the plant community and evenness of multifunctionality improved the model fit based on AIC. Modification indices were checked to explore, if additional paths, including direct effects from plant species richness to multifunctionality, would improve the model; this, however, was never the case. Furthermore, we used Pearson correlations to explore potential relationships among the different soil functions (*Figure 2*), and we identified the top performing plant species in monoculture for each response variable in all four environmental contexts (*Supplementary file 1*). GLMs were performed in SAS 9.3 (SAS Institute), correlations were run in STATISTICA 10 (Statsoft), and SEM was performed using Amos 5 (Amos Development Corporation, Crawfordville, FL).

## Data accessibility

Data are provided in *Supplementary file 3*.

## Acknowledgements

NE gratefully acknowledges funding by the DFG (Ei 862/2). NE, JH and AV acknowledge funding by the DFG in the framework of the Jena Experiment (FOR 1451). This project also received support from the European Research Council (ERC) under the European Union's Horizon 2020 research and innovation program (grant agreement no 677232 to NE). Further support came from the German Centre for Integrative Biodiversity Research (iDiv) Halle-Jena-Leipzig, funded by the DFG (FZT 118). BioCON was supported the NSF Long-Term Ecological Research (DEB-1234162), Long-Term Research in Environmental Biology (DEB-1242531), and Ecosystem Sciences (DEB-1120064) programs. MR acknowledges support from the ERC Advanced Grant 'Gradual Change'.

## Additional information

### Funding

| Funder | Grant reference number | Author |
|---|---|---|
| Deutsche Forschungsgemeinschaft | Ei 862/2 | Nico Eisenhauer |
| Deutsche Forschungsgemeinschaft | FZT 118 | Nico Eisenhauer<br>Jes Hines<br>Alfred Lochner |

| Deutsche Forschungsge-meinschaft | FOR 1451 | Nico Eisenhauer Jes Hines Anja Vogel |
| --- | --- | --- |
| European Research Council | ERC award no 677232 | Nico Eisenhauer |
| National Science Foundation | DEB-1234162 | Peter B Reich |
| National Science Foundation | DEB-1120064 | Peter B Reich |
| National Science Foundation | DEB-1242531 | Peter B Reich |

The funders had no role in study design, data collection and interpretation, or the decision to submit the work for publication.

## Author contributions
Nico Eisenhauer, Conceptualization, Resources, Data curation, Formal analysis, Supervision, Funding acquisition, Investigation, Visualization, Writing—original draft, Project administration; Jes Hines, Conceptualization, Formal analysis, Investigation, Visualization, Writing—original draft; Forest Isbell, Fons van der Plas, Sarah E Hobbie, Matthias C Rillig, Writing—review and editing; Clare E Kazanski, Anika Lehmann, Mengyun Liu, Alfred Lochner, Kally Worm, Investigation, Writing—review and editing; Anja Vogel, Data curation, Formal analysis, Writing—review and editing; Peter B Reich, Resources, Supervision, Investigation, Writing—review and editing

## Author ORCIDs
Nico Eisenhauer http://orcid.org/0000-0002-0371-6720
Clare E Kazanski http://orcid.org/0000-0001-7432-5666

## Decision letter and Author response
Decision letter https://doi.org/10.7554/eLife.41228.019
Author response https://doi.org/10.7554/eLife.41228.020

## Additional files
### Supplementary files
• Supplementary file 1. Supplementary Table S1.
DOI: https://doi.org/10.7554/eLife.41228.014
• Supplementary file 2. Supplementary Document A1.
DOI: https://doi.org/10.7554/eLife.41228.015
• Supplementary file 3. Supplementary Table S2.
DOI: https://doi.org/10.7554/eLife.41228.016
• Transparent reporting form
DOI: https://doi.org/10.7554/eLife.41228.017

### Data availability
All data generated or analyzed during this study are available in Supplementary File 3.

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
