## [Decision Letter]

Thank you for submitting your article "Plant diversity maintains multiple soil functions in future environments" for consideration by *eLife*. Your article has been reviewed by three peer reviewers, including Bernhard Schmid as the Reviewing Editor and Reviewer #1, and the evaluation has been overseen by Ian Baldwin as the Senior Editor.

The reviewers have discussed the reviews with one another and the Reviewing Editor has drafted this decision to help you prepare a revised submission.

Summary:

This paper takes the showcasing of diversity effects one step further by revisiting the BioCON experiment 17 years after establishment and focusing attention on soil multifunctionality. The effects of biodiversity is a general topic that has been very heavily studied; therefore, it is useful to ask what unique contributions this study makes. The short answer is, I believe, there are some very interesting and important findings that come out of this combination of careful scrutiny and determined experimentation.

The paper demonstrates that positive effects of plant diversity on soil ecosystem services are maintained under increased CO_2_ or nitrogen levels that are predicted to represent future environmental conditions. This novel finding is remarkable because there have been claims that higher resource levels may alter biodiversity-ecosystem functioning relationships, e.g. due to increased competition and dominance of high-performing species. However, although the authors identified higher evenness at higher diversity as main a cause of positive biodiversity effects, the effects were the same under the different global-change treatments. These treatments, which reflected realistic scenarios for the coming decades, had weaker effects than expected and than diversity variables (richness, evenness) themselves.

The paper is not only unique for the above results but also because these have been obtained in one of the longest-running ecological experiments and because multiple ecosystem functions are analyzed in univariate and multivariate fashion. The societal and policy implications are therefore highly relevant.

Essential revisions:

There are three main points that the reviewers identified. The first concerns the selection of dependent variables, the second the evenness measure and potential mechanistic explanations via functional-trait evenness and the third the statistical analysis without consideration of some random-effects terms.

Selection of dependent variables

– You analyze 4 metrics, the selection of these appears to have been a bit guided by what was easily measurable or had not been analyzed previously. For example, you refer a lot to carbon sequestration, but why wasn't soil carbon measured and reported then? In a sandy soil, after 17 years, there would have been a chance to see effects. With the current variable selection the paper is about diversity-effects on soil microbial respiration (basal and substrate-activated), root biomass, and soil aggregate stability. These variables are quite related and thus it may not be appropriate/surprising to combine them to multifunctionality/find that this "multifunctionality" parallels the single functions. The result may be different for another suite of functions. This probably always is the case, but likely less so with a larger and more diversified set of functions. Please consider to include further variables to obtain a more "designed" multifunctionality measure of soil ecosystem services.

Evenness measure and potential mechanistic explanations via functional-trait evenness

– The Shannon evenness (J') is intrinsically autocorrelated with species richness (see Smith and Wilson, 1996, for a discussion of the properties of this index). Thus it may not be totally fair to say richness worked via evenness. It may be better to say it was a combination of both or to use/add a richness-independent evenness measure to the analysis. Apart from this, it would be interesting to know if it was the evenness of functional traits or functional groups of plants that was causing the increased ecosystem functioning. You most likely would have the corresponding measures to enrich your analysis in this regard. Often it is hoped that measures of functional diversity may explain additional variation among communities with different species compositions. Even if this was not the case here, it would be useful to mention it, i.e. to say that functional evenness did not yield better mechanistic explanations (perhaps with results in supplement).

Statistical analysis without consideration of some random-effects terms

– Not much detail is given regarding the statistical analysis, in particular F-tests for significance. This is not too bad because you are more interested in effect sizes than significance levels, but still it would be useful to know if/why planted species richness was not tested against species composition and CO_2_ was not tested against Ring(CO_2_). Similarly, species richness interactions might have been tested against corresponding species composition interactions. If species richness was not tested in such a way, you might find it useful to refer to Schmid et al. (2017) where we suggest that sometimes this can be ok. For CO_2_, there was obviously less statistical power to find effects, because these had to be tested against variation between rings. This creates a caveat regarding the comparison of CO_2_ with species richness as effects. Please provide a more detailed description of the analyses, and report nominator and denominator degrees of freedom with their F values.

References:

Smith B., Wilson J.B. A Consumer's Guide to Evenness Indices. *Oikos*. Vol. 76, No. 1 (May, 1996), pp. 70-82. DOI: 10.2307/3545749

---

## [Author Response]

Essential revisions:There are three main points that the reviewers identified. The first concerns the selection of dependent variables, the second the evenness measure and potential mechanistic explanations via functional-trait evenness and the third the statistical analysis without consideration of some random-effects terms.Selection of dependent variables– You analyze 4 metrics, the selection of these appears to have been a bit guided by what was easily measurable or had not been analyzed previously. For example, you refer a lot to carbon sequestration, but why wasn't soil carbon measured and reported then? In a sandy soil, after 17 years, there would have been a chance to see effects. With the current variable selection the paper is about diversity-effects on soil microbial respiration (basal and substrate-activated), root biomass, and soil aggregate stability. These variables are quite related and thus it may not be appropriate/surprising to combine them to multifunctionality/find that this "multifunctionality" parallels the single functions. The result may be different for another suite of functions. This probably always is the case, but likely less so with a larger and more diversified set of functions. Please consider to include further variables to obtain a more "designed" multifunctionality measure of soil ecosystem services.

We acknowledge this critical comment that helped us to better justify our approach in the revised version of our manuscript. Although we acknowledge that this was previously not sufficiently clarified, the selection of soil variables was very carefully planned and a major logistic effort. We chose to use soil variables that (i) do not correlate very closely to each other (see new Figure 2) and (ii) represent key soil-related services (as outlined in the manuscript). We did not include soil carbon concentrations because we know about the tight correlation with soil microbial biomass carbon (e.g., Eisenhauer et al., 2010; Lange et al., 2015). Accordingly, in the revised version of our manuscript, we present the correlations among soil functions in new Figure 2 and provide a more detailed explanation of the selection of soil functions:

“The focal soil functions were carefully chosen to (i) represent different soil-related ecosystem services and (ii) to not be too tightly correlated with each other. […] Given the tight correlation between soil carbon concentrations and soil microbial biomass C (Lange et al., 2015; Eisenhauer et al., 2010), and because soil microbial biomass has been shown to be a significant predictor of many soil functions (see above), we focused on the latter in the present study.”

Evenness measure and potential mechanistic explanations via functional-trait evenness– The Shannon evenness (J') is intrinsically autocorrelated with species richness (see Smith and Wilson, 1996, for a discussion of the properties of this index). Thus it may not be totally fair to say richness worked via evenness. It may be better to say it was a combination of both or to use/add a richness-independent evenness measure to the analysis. Apart from this, it would be interesting to know if it was the evenness of functional traits or functional groups of plants that was causing the increased ecosystem functioning. You most likely would have the corresponding measures to enrich your analysis in this regard. Often it is hoped that measures of functional diversity may explain additional variation among communities with different species compositions. Even if this was not the case here, it would be useful to mention it, i.e. to say that functional evenness did not yield better mechanistic explanations (perhaps with results in supplement).

We appreciate this constructive comment. Accordingly, we calculated different functional diversity and evenness indices based on plot-level cover data (the latter to also address another comment below) in Table 2 and Figure 4—figure supplements 1-5. These additional analyses provided novel insights, and we can now show that, indeed, as suggested by the reviewer, functional diversity of the plant community was an important predictor of ecosystem multifunctionality (see new SEM in Figure 5).

Statistical analysis without consideration of some random-effects terms– Not much detail is given regarding the statistical analysis, in particular F-tests for significance. This is not too bad because you are more interested in effect sizes than significance levels, but still it would be useful to know if/why planted species richness was not tested against species composition and CO_2_ was not tested against Ring(CO_2_).

We apologize for the unclear explanation, and take two steps to more fully describe the statistics. First, we provide the SAS code as Supplementary file 2. Second, we added a sentence clearly stating that the effect of CO_2_ was, in fact, tested against the random effect of ring nested within CO_2_ (Supplementary file 2; Reich et al., 2001).

Similarly, species richness interactions might have been tested against corresponding species composition interactions. If species richness was not tested in such a way, you might find it useful to refer to Schmid et al. (2017) where we suggest that sometimes this can be ok.

We agree with the reviewer and now provide a justification of our approach. In fact, we followed the rationale by Schmid et al. (2017) and did not test plant species richness effects against species composition, because of changes in plant species richness and composition over time, partly due to treatment effects (details provided in Table 2). Thus, we were not able to test for interactive effects of species richness and environmental change drivers against species composition effects (Schmid et al., 2017). The respective text in the revised manuscript reads: “Moreover, we were not able to test for interactive effects of plant species richness and environmental change drivers against species composition effects, because species composition changed significantly over time in response to the experimental treatments (Table 2; Schmid et al., 2017).”

For CO_2_, there was obviously less statistical power to find effects, because these had to be tested against variation between rings. This creates a caveat regarding the comparison of CO_2_ with species richness as effects.

Agreed – thank you! This caveat was added to the statistical analyses.

Please provide a more detailed description of the analyses, and report nominator and denominator degrees of freedom with their F values.

Done – all details are provided in Tables 1 and 2.